# Protective role of RIPK1 scaffolding against HDV-induced hepatocyte cell death and the significance of cytokines in mice

Gracián Camps, Sheila Maestro, Laura Torella, Diego Herrero, Carla Usai[¤a], Martin Bilbao-Arribas , Ana Aldaz, Cristina Olagüe, Africa Vales, Lester Suárez-Amarán[¤b], Rafael Aldabe*, Gloria Gonzalez-Aseguinolaza *

DNA & RNA Medicine Division, CIMA, University of Navarra, Instituto de Investigación Sanitaria de Navarra, IdisNA, Pamplona, Spain

¤a Current address: Unitat mixta d'Investigació IRTA-UAB en Sanitat Animal, Centre de Recerca en Sanitat Animal (CReSA), Campus de la Universitat Autonoma de Barcelona (UAB), Bellaterra, Spain
¤b Current address: Asklepios BioPharmaceutical, Inc., Research Triangle Park, North Carolina, United States of America
* raldabe@unav.es (RA); ggasegui@unav.es (GG-A)

**Data Availability Statement:** Individual data from Fig 1 and the analytical quantification of the images in Figs 1, S1, and S2 have been included in S1 to S4. Raw data from the graphs shown in Figs 2, 3, 4

## Abstract

Hepatitis delta virus (HDV) infection represents the most severe form of human viral hepatitis; however, the mechanisms underlying its pathology remain incompletely understood. We recently developed an HDV mouse model by injecting adeno-associated viral vectors (AAV) containing replication-competent HBV and HDV genomes. This model replicates many features of human infection, including liver injury. Notably, the extent of liver damage can be diminished with anti-TNF-α treatment. Here, we found that TNF-α is mainly produced by macrophages. Downstream of the TNF-α receptor (TNFR), the receptor-interacting serine/threonine-protein kinase 1 (RIPK1) serves as a cell fate regulator, playing roles in both cell survival and death pathways. In this study, we explored the function of RIPK1 and other host factors in HDV-induced cell death. We determined that the scaffolding function of RIPK1, and not its kinase activity, offers partial protection against HDV-induced apoptosis. A reduction in RIPK1 expression in hepatocytes through CRISPR-Cas9-mediated gene editing significantly intensifies HDV-induced damage. Contrary to our expectations, the protective effect of RIPK1 was not linked to TNF-α or macrophage activation, as their absence did not alter the extent of damage. Intriguingly, in the absence of RIPK1, macrophages confer a protective role. However, in animals unresponsive to type-I IFNs, RIPK1 downregulation did not exacerbate the damage, suggesting RIPK1's role in shielding hepatocytes from type-I IFN-induced cell death. Interestingly, while the damage extent is similar between IFNα/βR KO and wild type mice in terms of transaminase elevation, their cell death mechanisms differ. In conclusion, our findings reveal that HDV-induced type-I IFN production is central to inducing hepatocyte death, and RIPK1's scaffolding function offers protective benefits. Thus, type-I IFN together with TNF-α, contribute to HDV-induced liver damage. These insights may guide the development of novel therapeutic strategies to mitigate HDV-induced liver damage and halt disease progression.

and 5 are presented in an attached excel file as supporting information (S5 Table). The NGS analysis data used to determine the editing efficacy and genome modification pattern after RIPK1edit treatment can be found in the link: https://www.ncbi.nlm.nih.gov/bioproject/PRJNA1090862.

**Funding:** This research was funded by Proyectos I+D de Generación de Conocimiento, PID2021-124455OB-I00, RTI2018-101936-B-I00 MCIN/ AEI /10.13039/501100011033/ FEDER Una manera de hacer Europa to GGA. Spanish Network of Advanced Therapies TERAV Network supported by Instituto de Salud Carlos III (ISCIII) and Funded by the European Union – NextGenerationEU, Recovery, Transformation and Resilience Plan grant RD21/0017/0001 to GGA and MB. GC, CU and DH, were supported by FPI fellowships from the Spanish Ministry of Economy and Competitiveness, and SM and LT were supported by FIMA's AC fellowship. The funders had no role in study design, data collection and analysis, decision to publish, or preparation of the manuscript.

**Competing interests:** The authors have declared that no competing interests exist.

## Author summary

Hepatitis D is the most aggressive form of viral hepatitis. Our manuscript underscores the complexity of HDV-induced liver damage, where both viral and host factors play significant roles. Previously, we demonstrated that pharmacological inhibition of TNF-α reduced HDV-induced liver damage. This result was corroborated in the present study using TNF-α-deficient mice. Moreover, we reported that the expression of the HDV antigen might have a cytotoxic effect, and HDV replication induces a strong activation of the innate immune system, accompanied by a substantial production of IFN-β. In this study, we discovered that RIPK1, a molecule described as a cell fate modulator acting downstream of TNF-α, plays a protective role during HDV replication. Contrary to our expectations, neither TNF-α nor macrophages (the primary producers of TNF-α), contributed to this protective effect. Instead, it seems type I IFN was involved. Interestingly, the role of type I IFN in HBV-induced liver damage has recently been proposed. Furthermore, our data reveals that several mechanisms of hepatocyte death are at play simultaneously during HDV replication, with apoptosis being one of them. Additional studies are needed to identify other mechanisms involved. Finally, these findings suggest that therapies targeting TNF-α and type-I IFN, or those increasing RIPK1 levels, might be effective in preventing or treating HDV-induced liver damage.

## Introduction

Hepatitis D virus (HDV) requires hepatitis B virus (HBV) for completion of its life cycle [1]. Co-infection with HBV and HDV is the most severe form of viral hepatitis, characterized by an increased risk of fulminant hepatitis, cirrhosis, liver decompensation, and hepatocellular carcinoma compared to HBV mono-infection [2]. It is estimated to affect 15–20 million individuals globally [3,4]. Despite the significant burden of this disease, the mechanisms underlying HDV-induced hepatocyte death and liver inflammation that conditions the severity of the disease and the rapid development of fibrosis are not fully understood [5]. HDV has previously been reported as not being directly cytotoxic to human hepatocytes. However, early studies showed that the expression of hepatitis D antigen (HDAg) resulted in significant cytotoxic changes in these cells [6]. We recently corroborated those results in mice overexpressing HDAgs in the liver [7]. Although this can be seen as an artificial situation that doesn't mirror the circumstances of natural viral infection, Ricco et al. found an association between HDV viral load and the severity of liver injury [8]. This brings up questions regarding the contribution of viral components to HDV-related pathology. Conversely, the role of the immune system in HDV pathogenesis is debated. Some studies emphasize the significance of HDV-specific adaptive immune responses [9,10], while others report weak or undetectable HDV-specific T cell responses. Moreover, there appears to be no correlation between the magnitude of HDV-specific CD4+ or CD8+ T cell responses and the clinical status of patients [11]. Interestingly, a recent study by Kefalakes et al. found a notable correlation between the size of both HDV-specific and non-specific CD8+ T cells expressing innate-like NKp30 (Natural Killer protein 30) and NKG2D (NK group 2D) receptors and the level of liver inflammation in HDV patients [12]. Furthermore, in vitro data showed that HDV infection of hepatic cell lines enhanced the cytotoxicity of CD8+ T cells. This effect was independent of MHC class I-TCR interaction, but it did require activation of the innate immune response and type I interferon

response [13]. Taken together, these results suggest that bystander activation of immune cells with cytotoxic capacity may contribute to liver pathology during HDV infection.

The lack of mouse models that accurately mimic human HDV infection has hindered progress in understanding the mechanisms underlying HDV-mediated liver damage. The available models, including immunodeficient mice with humanized liver or immunocompetent transgenic mice expressing the human sodium taurocholate cotransporting polypeptide (hNTCP) (receptor for HBV and HDV), do not fully replicate the characteristics of human HDV infection, particularly in terms of the induction of liver damage [14,15]. Recently, a HBV/HDV mouse model was developed using recombinant adenoassociated vectors (rAAV) to deliver HBV and HDV replication-competent genomes [7,16]. Interestingly, once HDV replication is established AAV genomes are hardly detectable, indicating that the rAAV acts as a HDV shuttle and is not required for HDV replication maintenance [16]. This model reproduces many of the features of HDV infection in humans, including liver inflammation and injury. Importantly, we found that liver injury can be partially ameliorated by TNF-α inhibition [7] which is in line with previous reports showing that TNF-α levels are higher in HDV-infected patients and that there is a close correlation between TNF-α and the severity of the disease [17]. TNF-α is an inflammatory cytokine recognized by TNF-α receptor 1 (TNFR1) on the cell surface, and downstream, the receptor-interacting serine/threonine-protein kinase 1 (RIPK1) acts as a cell fate regulator involved in both cell survival and cell death pathways [18]. The kinase activity of RIPK1 triggers cell death by apoptosis or necroptosis through the activation of caspase 8 or RIPK3-Mixed lineage kinase domain like protein (MLKL), respectively [19]. In line with this, necrostatin-1 (Nec1), an inhibitor of RIPK1 kinase activity. has been shown to have a protective effect in several models of acute liver injury, such as acetaminophen- and concanavalin-A-induced hepatocyte death [20,21]. Nec-1 was developed through a process of chemical screening and optimization aimed at identifying small molecule inhibitors of programmed cell death, particularly necroptosis. It functions by inhibiting RIPK1-induced necroptosis, thereby reducing MLKL activation [20]. However, RIPK1 may also play a protective role in hepatocytes [22–25]. In mice challenged with hepatotoxic agents such as lipopolysaccharides (LPS), poly(I:C) or murine hepatitis virus type 3 (MHV3), the absence of RIPK1 resulted in significant elevation of liver transaminases and cell death [22–25]. Furthermore, it was demonstrated that the scaffolding function of RIPK1, but not its kinase activity, was responsible for this protective effect. Interestingly, the inhibition of TNF-α or depletion of macrophages resulted in amelioration of liver injury associated with the absence of RIPK1 [22,23].

In our current study, we determined the main cellular source of TNF-α in HDV-replicating mouse livers and the role of RIPK1 in HDV-induced liver injury. We found that most of the TNF-α producing cells are macophages and the scaffolding function of RIPK1, but not its kinase activity, plays a protective role against HDV-induced liver damage. However, RIPK1 doesn't block cell death induced by TNF-α and/or macrophages. Instead, it appears to target apoptosis induced by type-I IFN. Additionally, we also found that in the absence of RIPK1, macrophage depletion resulted in severe liver damage that can be fatal.

## Results

### Macrophages are the main source of TNF-α in the liver of AAV-HBV/HDV mice

In our previous study, we demonstrated that TNF-α plays a role in HDV-induced liver damage and its pharmacological inhibition showed a partial protective effect [7]. In order to determine which cells are responsible for TNF-α production we used RNA *in situ* hybridization (ISH) RNAscope Fluorescent Multiplex Kit to analyze the localization of TNF-α mRNA and HDV

genomic (HDVg) and antigenomic RNA (HDVag) in the liver of 4 C57BL/6 wild type (wt) mice 21 days after AAV-HBV/HDV intravenous (iv) injection. Although in this analysis the HDVag was difficult to detect due to a weaker signal, we found a colocalization of HDVg and HDVag in the cells, indicating active replication of the virus (Fig 1A2). Interestingly we found that most cells expressing TNF-α mRNA were HDV RNA negative (indicated by orange arrows in Fig 1A1,1A2 and 1A3) and only a smaller percentage of TNF-α positive cells were HDV RNA positive (Fig 1A4). In fact, the automated quantification of the images revealed that over 80% of the cells expressing TNF-α were HDV negative (Fig 1B and S1 Table). Since HDV RNA expression and replication is restricted to hepatocytes thanks to the use of a hepatocyte specific promoter that controls the transcription of the HDVag, this result suggests that hepatocytes are not the main source of TNF-α. To confirm the hepatospecificity of HDV replication we performed a second ISH analysis in 2 of the animals using HDVg and HDVag probes combined with a probe to detect albumin as a hepatocyte marker. As shown in S1A Fig and S2 Table, the majority of HDV signal is detected in hepatocytes. Furthermore, in nearly all the HDVag positive hepatocytes HDVg is detected (S2 Table).

Previous studies have shown that liver macrophages are responsible for TNF-α production after various stimuli [19,20]. To analyze the role of this population, we combined TNF-α and HDV genome ISH with immunohistochemistry staining for the macrophage marker F4/80 using the same liver samples. As shown in Fig 1C and 1D and S3 Table, we found that most cells producing TNF-α are macrophages (colocalization of the red and yellow signal Fig 1C Merge). This finding was confirmed by imaging scanning and automated signal quantification (Fig 1D). Interestingly, 5 to 10% of macrophages expressing TNF-α also contain HDV RNA, and all the macrophages positive for HDV are positive for TNF-α (white arrow in Fig 1C). Then, we found 20–25% of cells expressing TNF-α that were F4/80 and HDV negative and a small number of HDV positive hepatocytes (Fig 1D). To further corroborate the presence of HDV in macrophages, ISH analysis was repeated but in this case the anti-F4/80 antibody was substituted by an anti-F4/80 RNA probe, and was combined with the anti-albumin and HDVg RNA probes (the anti-TNF-α was removed). As shown in S2A Fig and S4 Table, we found that 7 to 9% of the macrophages showed HDV positive signal. As expected the majority of HDV positive cells are hepatocytes, in fact from 22 to 29% of the hepatocytes were HDV positive (S4 Table).

## RIPK1 scaffolding function, but not its kinase activity, is critical for the survival of HDV-infected hepatocytes

Several studies have shown the key role of RIPK1 in maintaining liver homeostasis under conditions of macrophage activation and TNF-α production [22–25]. To assess the role of RIPK1 in HDV-mediated liver injury, we developed a CRISPR/Cas9 gene editing system that specifically knockdown RIPK1 expression in hepatocytes. We achieved this by expressing the Cas9 protein under the control of the human thyroxine binding globulin (TBG) liver specific promoter (S3A Fig) that was delivered using a rAAV vector with hepatic tropism. Two different guide (g) RNAs were designed (RIPK1g1 and RIPK1g2) and their editing efficacy was tested in vivo by analyzing RIPK1 protein expression in liver extracts by western blot. As shown in S3B Fig, RIPK1 protein expression was significantly reduced with RIPK1g2 and this guide was selected for further studies (the AAV vector containing Cas9 and RIPK1g2 was named RIPK1edit). In order to better characterize genome editing efficiency and variant distribution in RIP1Kedit treated livers, the RIPK1 targeted locus was amplified by PCR to generate barcoded libraries that were analyzed by next generation sequencing (NGS). We detect an indel average of 50% (S3C Fig) being the most common mutations small indels under 10 bp (S3D Fig).

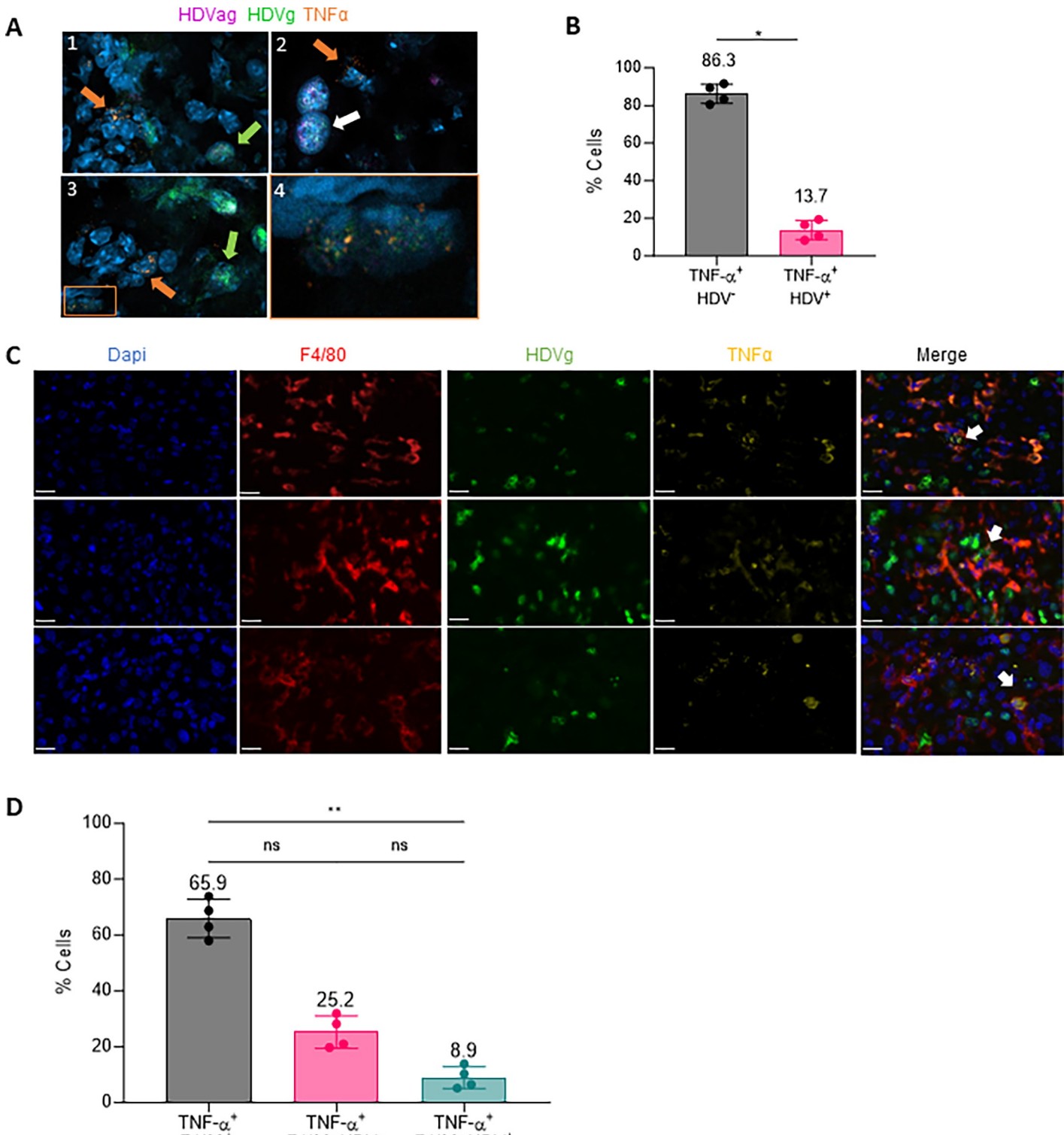

**Fig 1.** Macrophages are the main source of TNF-α production in AAV-HBV/HDV mouse livers. (A) TNF-α mRNA (orange) and HDV RNA genome (HDVg in green) and antigenome (HDVag in pink) distribution was analyzed by in situ hybridization (ISH) in the liver of C57BL/6 mice 21 days after receiving adenoassociated viral (AAV) vectors delivering both HBV and HDV genomes (HBV/HDV) at a dose of $5x10^{10}$ vg/mouse. Images show cells expressing TNF-α (orange arrows), cells positive for HDVg (green arrows) and cells positive for both HDVg and HDVag (white arrows). Panel 4 shows a magnification of a HDV positive cell expressing TNF-α. (B) The percentage of TNF-α+ HDV+ and TNF-α+ HDV- was quantified using image J (mean ± SD, n = 4). Statistical differences were determined by Mann-Whitney test p <0.05 (*). (C) Liver sections were analyzed for TNF-α mRNA (yellow) and HDVg (green) expression by in situ hybridization (ISH) followed by

immunofluorescence (IF) to detect F4/80 protein (red). Representative images of hybridized liver sections taken with the Vectra Polaris Automated Imaging System are depicted in; white arrows pointed to macrophages expressing TNF-α and that contain HDV RNA. The white bar in each image is 20μm long. (D) The percentage of TNF-α+ F4/80+, TNF-α+ F4/80+ HDV-, TNF-α+ F4/80- HDV+ and TNF-α+ F4/80+ HDV+ cells was quantified as described in methods. Individual data points and mean values ± standard deviations are plotted. Statistical differences were determined by Kruskal-Wallis test following Dunn's multiple comparison test p <0.05 (*).

These modifications introduced frameshift mutations, which explained the reduction in RIPK1 protein levels.

To determine the functional effects of RIPK1 downregulation, wt mice were treated with RIPK1edit and their susceptibility to LPS-induced liver injury and apoptotic cell death, which have been shown to be exacerbated in the absence of RIPK1 [22], was evaluated as described in S3E Fig. We found that RIPK1edit-treated animals showed a significantly higher susceptibility to LPS-induced liver damage compared to control animals, as shown by higher levels of transaminases and higher number of apoptotic cells identified by the expression of activated Caspase-3 (a-Casp3) (S3F–S3H Fig), demonstrating the functional impact of RIPK1edit in the liver of mice.

To investigate the role of RIPK1 in HDV-induced liver injury, wt mice were administered with RIPK1edit or an AAV vector expressing Cas9 but without guide (control). Thirty hours later (before the development of anti-AAV neutralizing antibodies that would block the transduction of a second AAV administration), each group was divided into two subgroups that received AAV-HBV/HDV (HBV/HDV) or AAV-HBV (HBV), respectively (Fig 2A). Transaminase levels were evaluated weekly up to day 21, when the animals were sacrificed (Fig 2A). As expected from our previous data [7,16] no liver damage was observed in control mice receiving HBV while a significant increase in transaminase levels was observed at day 14 and 21 in control HBV/HDV animals (Fig 2B). RIPK1edit has no effect on HBV-induced hepatocyte damage; however, it exerts a significant impact on HBV/HDV-induced liver injury, as evidenced by significantly higher transaminase levels in RIPK1edit-HBV/HDV mice compared to control-HBV/HDV mice at 14 and 21 days post-infection (dpi) (Fig 2B). At sacrifice, the expression of a-Casp3 and HDV antigen (HDAg) was analyzed by IHC and HDV genome levels were measured by RT-qPCR. We observed a higher number of a-Casp3 positive (apoptotic) hepatocytes in RIPK1-edit mice receiving HBV/HDV compared to control HBV/HDV mice; while no apoptotic cells were detected in the two groups of mice receiving HBV alone (Fig 2C). Moreover, there was a direct correlation between liver damage and the number of apoptotic hepatocytes in the two groups of mice receiving HBV/HDV (S4A Fig). Interestingly, while there were no major differences in HDV viremia (HDV in circulation) between control and RIPK1edit mice (Fig 2D), we observed that the number of cells expressing HDAg (Fig 2E and 2F) and the HDV genome levels (Fig 2G) at day 21 were significantly lower in RIPK1edit mice, likely due to increased hepatocyte apoptosis.

The experiment was repeated in C57BL/6 Rag1 deficient mice to evaluate the potential role of the adaptive immune system in the exacerbation of liver damage upon RIPK1 downregulation. As shown in S4B–S4D Fig, in the absence of B, T, and NKT cells, HBV/HDV-induced liver damage was also exacerbated after RIPK1 knockdown as shown by significant higher transaminase levels (S4B Fig) as well as a higher number of apoptotic hepatocytes (S4C Fig) and both parameters showed a direct correlation (S4D Fig).

To determine if the exacerbation of HBV/HDV-induced liver damage observed in RIPK1edit mice was related to RIPK1 kinase activity, HBV/HDV injected animals were treated daily with a dose of 2.5 mg/kg Nec1 (an inhibitor of RIPK1 kinase activity) and HBV injected mice were used as controls. All animals were bled weekly and sacrificed at 21 dpi. The efficacy of Nec-1 treatment was demonstrated by a notable reduction in MLKL expression levels in the

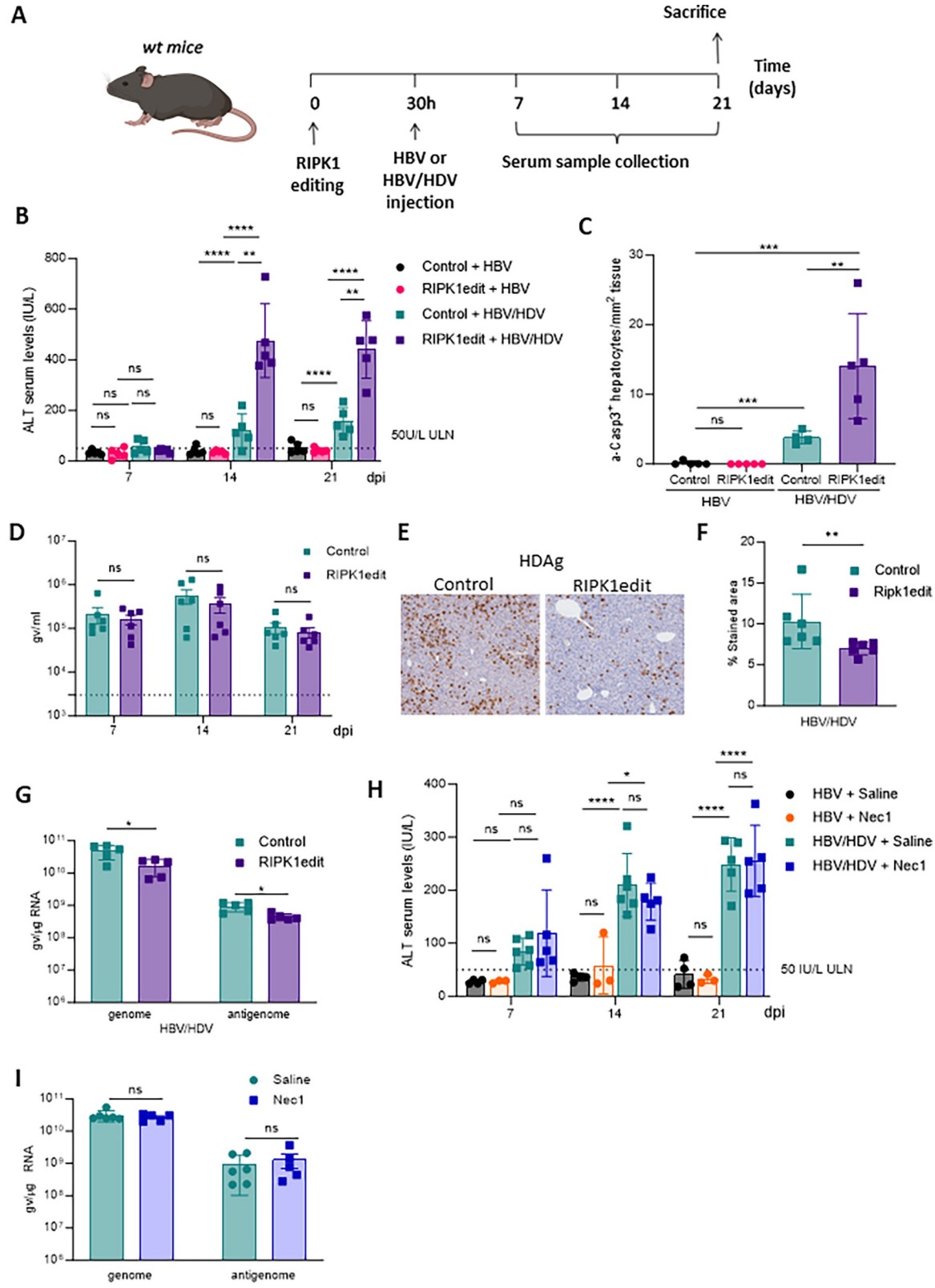

**Fig 2. RIPK1 scaffolding function partially prevents hepatocellular apoptosis in HBV/HDV mice.** (A) Schematic representation of the experimental procedure, 6/8-week-old C57BL/6 wt mice were intravenously injected with $10^{11}$ vg/mouse of AAV-SaCas9-RIPK1g2

(RIPK1edit) or an AAV expressing SaCas9 without guide (control) and after 30 hours they were administered with 5x10^10 vg/mouse AAV-HBV (HBV) or AAV-HBV/HDV (HBV/HDV). Peripheral blood was extracted at 7, 14 and 21 days post-injection (dpi). Liver damage was analyzed by (B) quantification of serum ALT levels (international units (IU)/L) and (C) quantification of a-Casp3 + hepatocytes/area after Immunohistochemistry (IHC). (D) HDV viremia in circulation by RT-PCR was determined in control and RIPK1edit mice 7, 14 and 21 dpi (E). IHC analysis against HDAgs was performed at 21 dpi in the liver sections of both groups and (F) quantified using Image J. (G) The presence of HDV genomes and antigenomes in mouse liver samples was quantified by RT-PCR. (H) Analysis of serum ALT levels in wt mice daily treated with Nec1 or saline was performed 7, 14 and 21 days after administration of HBV or HBV/HDV. (I) The presence of HDV genomes and antigenomes in mouse liver samples was quantified by RT-PCR. Individual data points and mean values ± standard deviations are shown. Statistical analysis was performed by one-way ANOVA followed by Bonferroni multiple-comparison test. $p < 0.05$ (*), $p < 0.01$ (**), $p < 0.001$ (***), $p < 0.0001$ (****) and ns = non-significant. IU: International units. (A) Created with BioRender.

liver of mice (S4E Fig). However, as depicted in Fig 2H, Nec-1 treatment had no discernible effect on HDV-induced liver damage, as evidenced by the absence of differences in transaminase increase between the two groups. Furthermore, no differences in HDV genome and antigenome levels were found between Nec-1 treated or untreated mice (Fig 2I). Thus, this result indicates that RIPK1 kinase activity appears to play no role in the protective effect of RIPK1 during HDV replication.

In summary, our data suggest that the scaffolding function of RIPK1 is critical for the survival of HDV-infected hepatocytes and that the increased liver damage in RIPK1-edited animals is independent of the adaptive immune response.

## Macrophages and TNF-α are not involved in the exacerbation of HDV-mediated liver damage in the absence of RIPK1

Previous studies have shown that, in the absence of RIPK1, liver damage induced by the administration of different agents to mice is attenuated by macrophage depletion or anti-TNF-α treatment [22,23]. Here, we examined the role of both in the increase of liver damage observed in HBV/HDV mice after RIPK1 downregulation. As a first step, we analyzed the presence of macrophages in the livers of mice 21 days after HBV/HDV or HBV injection in RIPK1edit and control mice by F4/80 IHC. We found a significant increase in liver macrophages in HBV/HDV mice in comparison to HBV mice, that were further increased when mice were pretreated with RIPK1edit (Fig 3A and 3B). In order to determine the role of the macrophage population in the exacerbation of liver damage due to the absence of RIPK1, they were removed using chlodronate-loaded liposomes (CLL) and liver injury was evaluated after HBV/HDV injection. CLL was administered iv 1 day before HBV or HBV/HDV injection and then every 4 days up to day 20 pi, as described in Fig 3C. Macrophage-depletion was confirmed by F4/80 IHC (S5A and S5B Fig). In control animals (administered with AAV expressing Cas9 but no guide) macrophage depletion resulted in a slight reduction of liver damage after HBV/HDV injection at 21 dpi (Fig 3D) and in a significant reduction of TNF-α expression (S5C Fig). However, in RIPK1edit the administration of CLL had a severe detrimental effect with a highly significant increase in transaminase levels in comparison to control mice (Fig 3D). Interestingly, the number of apoptotic hepatocytes in RIPK1-edited and CLL-treated mice was lower than in RIPK1-edited non CLL-treated mice (Fig 3E), and liver histology revealed the presence of necrotic areas (Fig 3F), indicating most likely the activation of a death mechanism other than apoptosis. The increment in liver damage correlates with a significant reduction in HDV genomes and antigenomes in the liver of RIPK1 edit CLL-treated animals (S5D Fig).

Next, we assessed the role of TNF-α in the increased liver damage in RIPK1-edited mice due to HBV/HDV by employing C57BL/6 mice lacking TNF-α. TNF-α KO mice receiving HBV/HDV showed lower transaminase levels than wt animals 21 dpi (Fig 3G), in line with our previous published data [7]. However, transaminase levels and the number of apoptotic

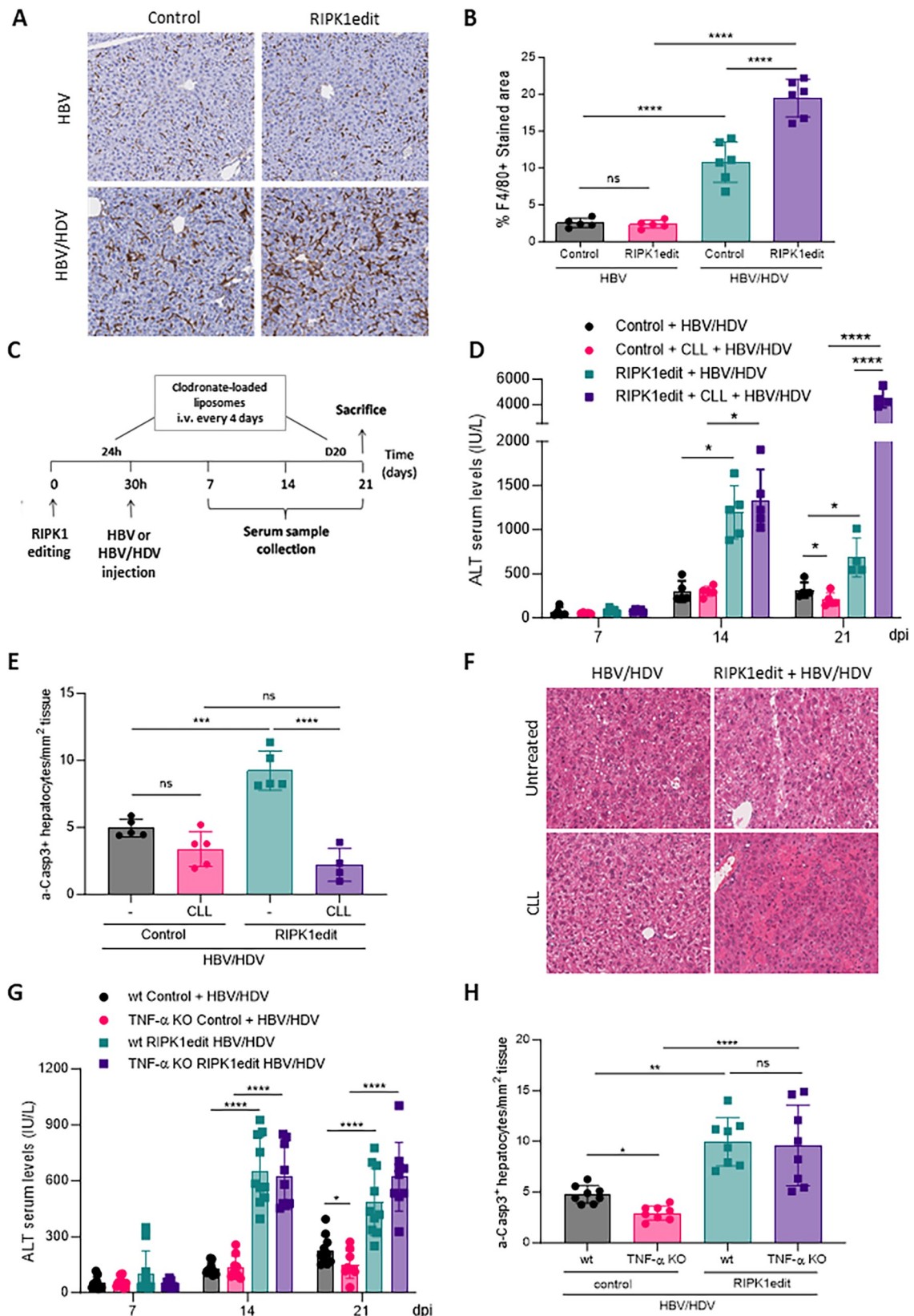

**Fig 3.** The exacerbation of liver damage in RIPK1-edited mice upon HBV/HDV co-injection is not mediated by macrophages or TNF-α. (A) The presence of macrophages in the livers of control and RIPK1edit wt mice was analyzed 21 dpi of HBV or AAV- HBV/HDV by F4/80 immunohistochemistry (IHC). Representative IHC images of F4/80-stained livers are shown (200x). (B) Quantification of F4/80 IHC stained area (mean ± SD). Significant differences were determined by one-way ANOVA followed by Bonferroni multiple-comparison test. (C) Schematic representation of the experimental procedure, 6/8-week-old C57BL/6 wt mice were treated as described in Fig 2A and for macrophage depletion, animals were iv injected with clodronate-loaded liposomes (CLL) every 4 days starting the day before HBV/HDV injection up to day 20. Liver damage was analyzed by (D) determination of serum ALT levels (IU/L) (E) quantification of a-Casp3+ hepatocytes/area (Individual data points and mean values ± SD are shown) and (F) liver histology analysis on H&E-stained sections, representative images are shown. Statistical differences were determined by two-way (D) or one-way (E) ANOVA followed by Bonferroni multiple-comparison test. (G-H) 6-8-week-old C57BL/6 wt and TNF-α KO mice were treated as described in Fig 2A. Liver damage was analyzed by (G) determination of serum ALT levels (IU/L) and (H) quantification a-Casp3+ hepatocytes. Individual data points and mean values ± standard deviations are shown. Statistical differences were determined by two-way (F) or one-way (G) ANOVA followed by Bonferroni multiple-comparison test. $p < 0.05$ (*), $p < 0.01$ (**), $p < 0.001$ (***), $p < 0.0001$ (****) and ns = non-significant.

hepatocytes in RIPK1-edited mice were similar in TNF-α KO and wt mice (Fig 3G and 3H), indicating that the exacerbation of liver damage observed in RIPK1edit mice is not related to TNF-α production. In accordance with these data a decrease in HDV genomes and antigenomes was observed RIPK1edit TNF-α KO as well as in wt mice (S5E Fig).

In conclusion, neither macrophages nor TNF-α play a major role in the increased liver damage observed in HBV/HDV-infected mice in the absence of RIPK1. Instead, our data suggest that macrophages play a protective role during HBV/HDV infection in RIPK1-knock-down mice, in which apoptotic cell signaling is intensified.

## HDAg-mediated hepatocyte apoptosis it is not affected by the absence of RIPK1

In our previous study we have shown that HDV small and large antigens (S-HDAg and L-HDAg), in particular the S-HDAg, are also involved in the induction of hepatocyte death [7]. In order to determine if RIPK1 plays a role in protecting hepatocytes form HDAg-induced cell death, RIPK1 edited animals received a dose of an AAV expressing S-HDAg and transaminase levels were analyzed 7 and 14 days after (the time points we have previously determined transaminase levels were elevated [7]). As shown in Fig 4A, no differences in transaminase levels were observed between control and RIPK1edit animals. Furthermore, the number of apoptotic cells although higher than in control animals were very low and not altered by RIPK1 editing (Fig 4B), suggesting that the main mechanism of S-HDAg-induced cell death is not apoptosis and RIPK1 played no role. Importantly, no major differences in the S-HDAg expression levels were observed between the two groups (Fig 4C).

## RIPK1 downregulation has no effect in mice lacking type I IFN receptor

We have previously shown that HDV replication, using the AAV-HBV/HDV animal model, leads to the production of high levels of type-I IFN via the MAVS signaling cascade [16]. This has also been confirmed *in vitro* using human cells [13,26]. We also reported no differences in the magnitude of HDV-induced liver damage between C57BL/6 mice lacking the IFNα/β receptor (IFNα/βR KO) and wt mice, thus we concluded that type I IFN played no role on HDV-induced liver damage [7]. However, we did not investigate the cell death mechanism in those animals.

Here to determine if in the absence of RIPK1, type-I IFN has a role in the exacerbation of liver damage induce by HBV/HDV replication, we treated IFNα/βR KO and wt mice with RIPK1edit followed by HBV/HDV. As shown in Fig 5, and as previously reported [7], there were no significant differences in transaminase levels between wt and IFNα/βR KO mice after HBV/HDV injection. However, while RIPK1 knockdown had a significant impact on the liver

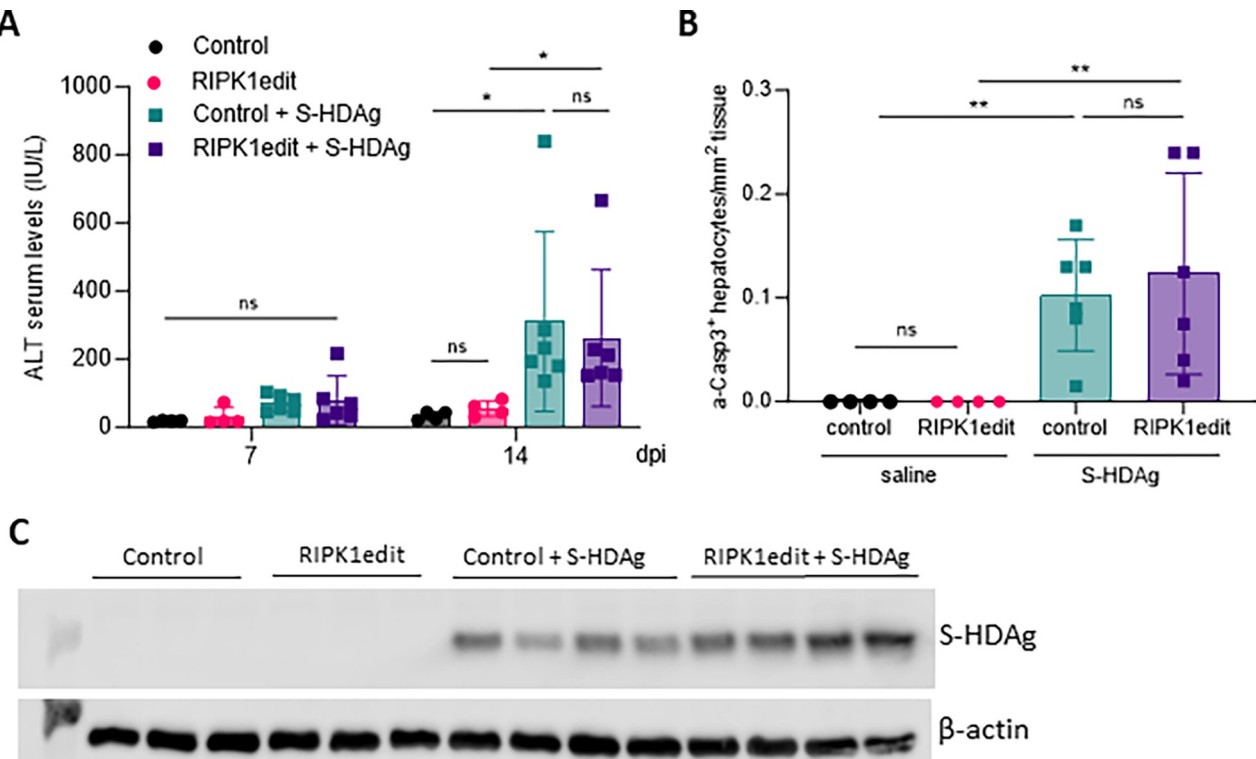

**Fig 4.** RIPK1 downregulation has no effect over HDAg-induced liver damage. 6/8-week-old C57BL/6 wt mice were injected with Cas9 control or RIPK1edit and 30 hours later both groups were injected with an AAV expressing HDV-Small Antigen (S-HDAg). Liver damage was analyzed by (A) determination of serum ALT levels (IU/L) at 7 and 14 dpi (dpi) and (B) quantification of a-Casp3+ hepatocytes/area at 14 dpi. (C) Western blot analysis to determine S-HDAg expression was performed at 14 dpi in the liver extracts. Individual data points and mean values ± standard deviations are shown in A and B. Statistical differences were determined by two-way ANOVA. $p < 0.05$ (*), $p < 0.01$ (**) and ns = non-significant.

damage in wt animals, there were no differences in transaminase levels in IFNα/βR mice regardless of RIPK1 expression (Fig 5A). Interestingly, the number of a-casp3 was lower in IFNα/βR than in wt, indicating differences in the mechanism of HDV-induced cell death in the absence of response to type-I IFN (Fig 5B). In addition, the analysis of HDAg expression and HDV genomes and antigenomes revealed that the number of positive cells and HDV replication levels were very similar in wt and IFNα/βR KO mice. However, whereas RIPK1 downregulation resulted in a decrease of HDAg expressing cells in wt mice, this decrease was not observed in IFNα/βR KO animals (Fig 5C and 5D), which correlates with the absence of disease exacerbation in these animals. The same was true for HDV genome and antigenome copy number (Fig 5E). Collectively, these results suggest that when response to type-I IFN is intact HDV-induces apoptotic cell death and RIPK1 plays a protective role but in its absence other cell death mechanisms seem to come into play.

## Discussion

Cell death plays a central role in the resolution and progression of disease. During viral infection, controlled cell death can eliminate pathogens, but it can also cause uncontrolled organ damage [27]. HDV infection is the most aggressive form of viral hepatitis infection, however, the mechanism involved in HDV-induced cellular death has not yet been fully understood. While apoptosis is a prevalent form of cell death in hepatocytes, other mechanisms such as necroptosis, ferroptosis, and autophagy-mediated cell death can also occur [28]. Liver

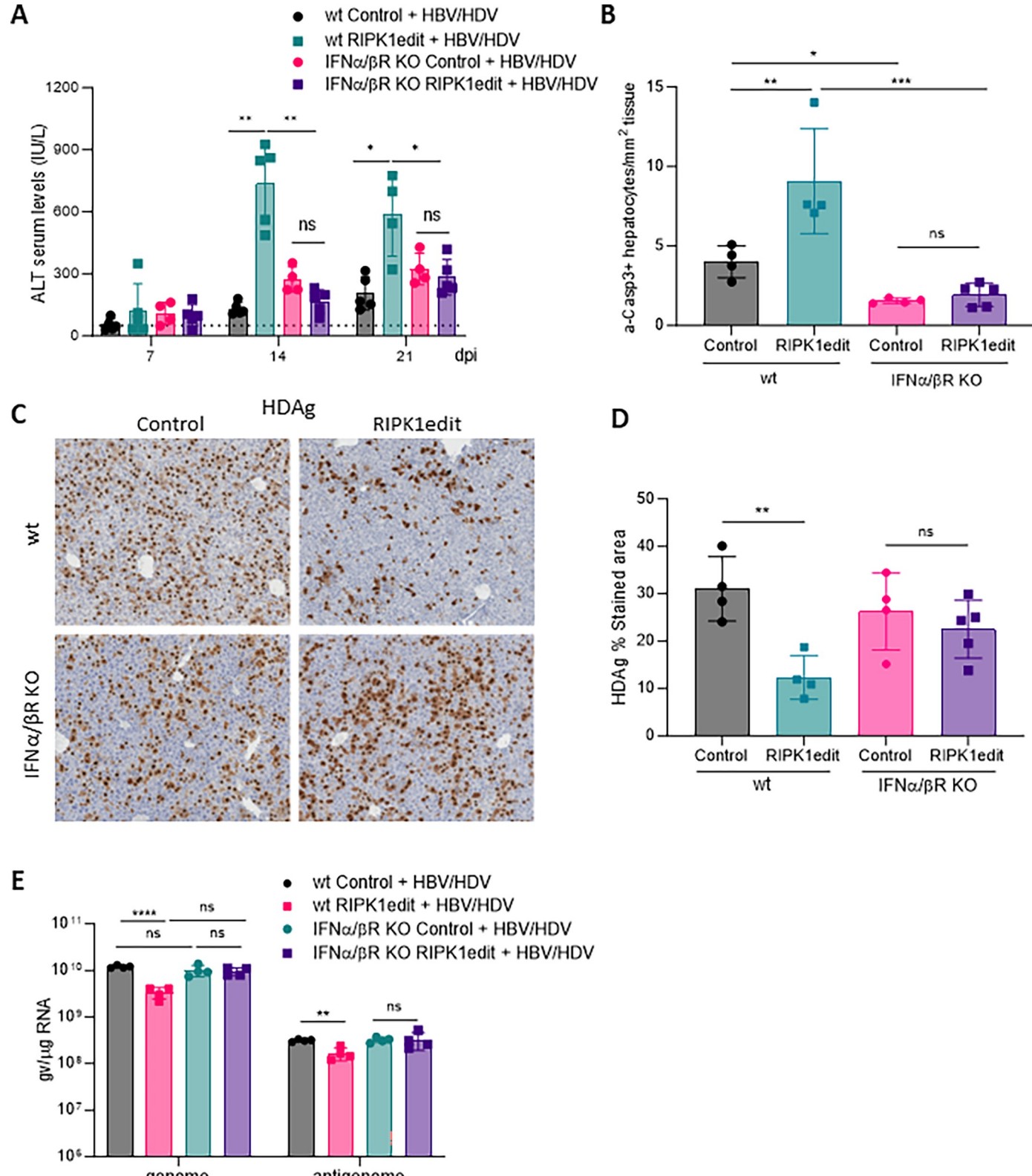

**Fig 5.** RIPK1 downregulation has no effect over HDV-induced liver damage in the absence of type-I IFN response. 6/8-week-old C57BL/6 wt and IFN-α/βR KO mice were treated as described in Fig 2A. Liver damage was analyzed by (A) determination of serum ALT levels (IU/L) at 7, 14, and 21 dpi and (B) quantification of a-Casp3+ hepatocytes/area at 21 dpi. (C) IHC against HDAgs was performed at 21 dpi in the liver sections of control and RIPK1edit mice injected with the HBV/

HDV vectors. (D) Quantification of HDAg expression in the aforementioned IHC images for each mouse. (E) The presence of HDV genomes and antigenomes in mouse liver samples was quantified by RT-PCR. Individual data points and mean values ± standard deviations are shown. Statistical differences were determined by two-way ANOVA (A and E) or one-way ANOVA (B and D) followed by Bonferroni multiple-comparison test. $p < 0.01$ (**), $p < 0.001$ (***), $p < 0.0001$ (****) and ns = non-significant.

parenchymal cells can activate different cell death pathways simultaneously, depending on the circumstances. The type of cell death that ultimately occurs is determined by factors such as the intensity and nature of the signal, the cellular context, and the presence of inhibitors or activators of specific cell death pathways [28]. Thus, understanding the complexity of the underlying mechanisms and the interplay between different pathways is crucial for developing effective strategies for treating liver diseases.

Previously, we showed that replication of HDV and HBV in mouse hepatocytes after AAV-mediated genome delivery induced significant liver damage, accompanied by the production of various cytokines including TNF-α and IFN-β [7]. In addition, we found that pharmacological inhibition of TNF-α resulted in a significant reduction in HDV-mediated liver damage [7]. These results have been corroborated in the present study using TNF-α deficient mice in which we observed a significant reduction, but not elimination, of HDV-induced liver injury. Here, we determined that TNF-α is primarily produced by activated macrophages. Consistent with this observation, depletion of macrophages significantly reduced TNF-α production and also partially attenuated HDV-induced liver damage. Our results suggest that HDV-induced hepatocyte-derived signals likely activate macrophages, which are responsible for producing this cytokine. Interestingly, we also found a non-negligible percentage of macrophages that were positive for HDV RNA and all of them expressed TNF-α. Since initial HDV RNA synthesis in our system is controlled by a liver-specific promoter, and mouse cells cannot be directly infected by circulating HDV infectious particles, this finding suggests that HDV RNA is transferred from hepatocytes to macrophages through an alternative mechanism and not by AAV-HDV-mediated transduction or HDV infection. It has recently been described that HDV can efficiently spread without envelopment through the proliferation of infected cells [29], so one possibility is that HDV-positive hepatocytes proliferate and spread HDV genomes to other cells, including macrophages. Another potential explanation is that macrophages uptake extracellular vesicles (EVs) produced by hepatocytes harboring HDV replication. The formation of EVs from HDV- and HBV-infected hepatocytes has been reported by other groups [30,31]. EVs can be taken up by monocytes, macrophages, and dendritic cells, inducing their activation [32]. However, given that the replication of HDV genomes is hepatocyte-independent [33], we cannot rule out the possibility that HDV replicates in macrophages, thereby directly activating the innate immune response in these cells.

Different studies have shown that TNF-α produced by liver macrophages plays a key role in both acute and chronic liver disease [22,23]. As described, TNF-α is recognized by TNFR1, which interacts with RIPK1, a protein with kinase activity that has emerged as a central molecular switch in controlling the balance between cell survival and cell death [18]. Depending on the type of insult, RIPK1 can promote cell survival, apoptosis, or necroptosis [19–25]. To study the potential role of RIPK1 in HDV-induced cell death, we developed a hepatocyte-specific gene-editing system using CRISPR-Cas9 delivered by an hepatotropic AAV vector. Using this in vivo editing system, we found that downregulation of RIPK1 expression resulted in a significant increase in liver damage after HBV/HDV injection that was not observed when HBV was administered alone. This suggests that RIPK1 plays a protective role during HDV infection. However, inhibiting RIPK1 kinase activity with Nec-1 showed that this protection is not related to its kinase activity. This suggests it is most likely related to its scaffolding function, as

previously demonstrated in other contexts [25]. Furthermore, the exacerbation of liver damage in the absence of RIPK1 was accompanied by an increase in the number of apoptotic cells. Although T cells have been suggested to be involved in HDV-induced liver damage [9,10], we found that immunodeficient mice lacking T cells had a similar magnitude of liver injury to wt mice and the downregulation of RIPK1 has a similar effect, indicating that other mechanisms independent of an uncontrolled T cell immune response against HDV-infected cells are involved in the pathology.

Interestingly, different studies have shown that the death of RIPK1-deficient hepatocytes induced by various agents (such as MHV3 infection, Poly(I:C), concanavalin A, LPS, or α-galactosylceramide) is mediated by TNF-α primarily produced by liver macrophages [20–25]. However, to our surprise, the absence of TNF-α or the elimination of macrophages did not alleviate HDV-induced liver damage in the absence of RIPK1. In fact, the depletion of liver macrophages had a clear detrimental impact. RIPK1 gene-edited mice in which macrophages were depleted showed a dramatic increase in hepatic injury after HBV/HDV administration, and the animals were nearly moribund at the end of the study. Histologically, we observed the formation of large necroinflammatory foci, which probably reflects that apoptotic hepatocytes were not removed by macrophages, a process that is crucial for the maintenance of liver health and homeostasis [34,35]. These results support the evaluation of macrophages as a potential cell-based therapy for the control of HDV-induced liver damage, a strategy that is currently being evaluated in clinical trials for the treatment of liver cirrhosis [36].

Previously, we and others have shown that HDAgs may play a role in HDV-induced liver damage [6,7]. Thus, we explored whether RIPK1 was involved in reducing the toxic effects produced by HDAg, but we observed no differences in the magnitude of liver damage induced by HDAg overexpression in the presence or absence of RIPK1.

Finally, we investigated the role of type-I IFN response in the exacerbation of liver damage in the absence of RIPK1. We have previously demonstrated in the HBV/HDV mouse model that HDV replication induced MAVS-mediated IFN-β production [16]. This result was corroborated in human cells showing that MDA5 was involved in the detection of HDV RNA which results in the production of type I IFN [14,26]. Interestingly, RIPK1 downregulation in IFNα/βR KO mice had no effect on the liver damage induced by HDV. Specifically, the magnitude of this damage, gauged by transaminase levels, remained consistent regardless of the presence or absence of RIPK1. Furthermore, in line with our previous results, we observed no differences in the severity of HDV-induced liver injury between wt and IFNα/βR KO mice. However, when examining the number of apoptotic cells, we found a significantly lower count in IFNα/βR KO mice compared to wt mice, regardless of the presence or absence of RIPK1. This suggests that in the absence of a response to type I IFN, the mechanism of HDV-induced cell death largely differs, though not entirely, from that in wt mice.

These results might suggest that in wt mice, type I IFN may be involved in HDV-induced apoptosis, and RIPK1 plays a protective role. Recently, it has been shown that type I IFN signaling induces cell death in HBV-hepatocytes by suppressing the unfolded protein response [37]. Furthermore, constitutive activation of MDA5 has been shown to lead to cell death [38]. Our previous data showed sustained activation MAVS-MDA5 activation in AAV-HDV mice leading to a sustained production of IFN-β. Thus it is not unlikely that type-I IFN is involved in HDV-induced liver damage [16]. Nonetheless, there is limited information regarding the interplay between type I IFN and RIPK1 and additional studies are needed.

Our findings might support the use of RIPK1 overexpression as a strategy to mitigate HDV-induced damage. However, the impact of RIPK1 overexpression may vary depending on the cellular context, influenced by factors such as the cellular environment, the interaction with other signaling molecules, and the nature of the cellular stress or insult [18,19]. While

excessive RIPK1 activity could lead to cell death and tissue damage in certain scenarios, it might promote cell survival and tissue equilibrium in others [18,19]. Further investigations are needed to clarify the specific role of RIPK1 overexpression in the AAV-HBV/HDV animal model.

In summary, our data indicate that HDV-induced cell death is complex, with both viral and host factors involved explaining why eliminating just one of these factors has only a partial effect on the extent of the damage. We found that type I IFN, along with TNF-α, are responsible for HDV-induced liver damage. It appears that various cell death modes coexist in the HDV-affected liver, a phenomenon recently termed "PANoptosis." PANoptosis describes a novel cell death routine where various cell death modes are engaged simultaneously, in some cases in response to pathogen's infection [39]. Remarkably, it has been shown that RIPK1 regulates Yersinia-Induced PANoptosis [40]. Further research is required to understand the interplay between different cell death pathways and the potential role of PANoptosis in HDV-induced liver disease. This knowledge will enhance our understanding of the intricacies and connections in HDV-induced pathogenesis and assist in the development of therapeutics.

## Material and methods

### Ethics statement

The animal experimental design was approved by the Ethics Committee for Animal Testing of the University of Navarra.

Samples from patients included in the study were provided by the Biobank of the University of Navarra and were processed following standard operating procedures approved by the Ethical and Scientific Committees (Reference: 2020–011). Formal written consent was obtained from the patients (all adults) and samples have been anonymized.

### Animals and treatments

C57BL/6 mice were purchased from Harlan Laboratories (Barcelona, Spain). Rag1- (Rag1 KO), IFNα/β receptor- (IFNα/βR KO) and TNF-α (TNF-α KO) deficient mice, all of them on a C57BL/6 background, were bred and maintained at the animal facility of Cima-Universidad de Navarra. Six- to eight-week-old male mice were used in all experiments. Mice were kept under controlled temperature, light, and pathogen-free conditions. Blood collection was performed by submandibular bleeding, and serum samples were obtained after centrifugation of total blood. Animals were euthanized by cervical dislocation after being anesthetized at the indicated time points. Liver samples were collected for histological and molecular analysis. The HBV and HBV/HDV mouse model were generated by the intravenous (iv) administration of AAV-HBV or AAV-HBV+AAV-HDV, respectively as previously described [16]. RIPK1 gene edition was performed by iv injection of AAV carrying liver specific CRISPR-Cas9 editing systems targeting *Ripk1* locus [41]. Lipopolysaccharide (LPS) (Sigma-Aldrich) was administered intraperitoneally (ip) at 0.4 μg/g of body weight, in a volume of 100 μl. Necrostatin-1 (Nec-1) was administered ip at a dose of 2,5 mg/kg. Macrophage depletion was achieved by iv administration of 100 μl clodronate-loaded liposomes (Clodlip BV).

### Multiplex fluorescence RNA in situ hybridization (ISH) and ISH in combination with F4/80 immunofluorescence (IF)

In situ Hybridization (ISH) was performed with the ACD RNAscope Fluorescent Multiplex Kit (Advanced Cell Diagnostics, USA) in liver sections. Firstly, livers were cryopreserved in an OCT cryomold and were cut in 10 μm-sections. Then, sections were fixed by incubating with

fresh 4% PFA for 30 minutes diluted in PBS. After that, samples were washed 2x with fresh PBS and dehydrated with EtOH. The EtOH solutions were prepared immediately before the dehydration and were diluted in Ambion DEPC-treated water (Invitrogen, #AM9920). When the EtOH was completely evaporated, a hydrophobic barrier was created with the Immedge hydrophobic barrier pen (Vector Laboratories, #H-4000). Then, samples were treated with RNAscope Protease IV for 30 minutes at RT. Next, sections were washed 2x with PBS and were hybridized in the HybEZ Oven for 2 hours at 40°C with the RNA probes from ACD: Mouse TNF-α Atto-550, HDV genome Alexa-Fluor 488, and HDV antigenome Atto-647 or Mm-Adgre1-C3 (F4/80), Mm-Alb-C1 (Alb) and V-HDV-GT1-O1- HDVg) or Mm-Alb-C1 (Alb), V-HDV-GT1-O1-C2 (HDVg), V-HDV-Antigenome-C3 (HDVag). After that, the slides were washed 2x with 1X Wash Buffer and were incubated with the RNAscope Detection Reagents to amplify the hybridization signals. Finally, the liver sections were incubated for 30 seconds with DAPI and mounted with ProLong Gold Antifade Mountant solution (Thermo Fisher, #P10144). For ISH combined with F4/80 IF HDV antigenome probe was removed, and after the last incubation with the RNAscope Detection Reagents, anti-F4/80 antibody was added diluted 1:10000 and incubated overnight at 4°C. Then slides were washed 3 times with PBS and incubated with anti-rat antibody for 30 min diluted 1:200, followed by an anti-rabbit Alexa 647 for 1h at 1:200. Finally, the liver sections were incubated for 30 seconds with DAPI and mounted with ProLong Gold Antifade Mountant solution (ThermoFisher, #P10144).

Fluorescent Images were acquired with the LSM 880 (Zeiss, Jena, Germany) laser-scanning confocal microscope and the Vectra Polaris Automated Quantitative Pathology Imaging System (Perkin Elmer). Classification and quantification of RNA+ cells types in 2D immunofluorescence confocal images has been carried out using a fully automated macro developed for Fiji/ImageJ, an opensource Java-based image processing software [42]. The macro was developed by the Imaging Platform at the Center for Applied Medical Research (CIMA). The Fiji/ImageJ macro was designed to classify cells types into RNA+ / RNA- Cells by colocalizing total Cells (DAPI), with the different fluorescent channels (specific markers) throughout whole tissue image. Each marker signal was segmented with intensity-based Otsu thresholding method to obtain a binary mask that represents as a defined ROIs the spatial localization of each cell type. For total cell segmentation the result mask from DAPI was post-processed using marker controlled watershed algorithm included in MorphoLibJ [43] to separate high cell density regions. Qualitative colocalization with RNA channel was computed with bitwise mask operations.

## CRISPR/Cas9 vectors and gRNA design

The pX602-AAV-TBG::NLS-SaCas9-NLS-HA-OLLAS-bGHpA;U6::BsaI-sgRNA plasmid that contains the Staphylococcus aureus Cas9 (SaCas9) expressed under the TBG promoter, the single guide RNA (sgRNA or gRNA) under U6 promoter and ITR sequences for AAV vector production was a gift from Feng Zhang (Addgene plasmid # 61593). gRNAs targeting exonic regions of the *Ripk1* gene (g1 and g2) were designed and selected using Benchling software (www.benchling.com). The selected 21-nt sequence 5'-AAGGGAACTATTCGCTGGTGA-3' was cloned upstream of the 5'-NNGRRT-3' PAM sequence of SaCas9. Annealed oligonucleotides coding for the guide RNA sequences (Sigma) were cloned into BsaI site of pX602 vector using standard molecular cloning techniques.

## Recombinant AAV production

The AAV genomes were packaged in AAV serotype 8-capsids (AAV8) as previously described [16]. Briefly, for each production, the AAV shuttle vector and the packaging plasmid pDP8.

ape (Plasmid factory) were co-transfected into HEK293T cells. The cells and supernatants were harvested 72 h upon transfection and virus was released from the cells by three rounds of freeze–thawing. Crude lysate from all batches was then treated with DNAse and RNAse for 1 h at 37˚C and then kept at -80˚C until purification. Purification of crude lysate was performed by ultracentrifugation in Optiprep Density Gradient Medium-Iodixanol (SigmaAldrich). Thereafter, iodioxanol was removed and the batches concentrated by passage through Amicon Ultra-15 tubes (Ultracel-100K; Merck Millipore). For virus titration, viral DNA was isolated using the High Pure Viral Nucleic Acid kit (Roche Applied Science). Viral titers in terms of viral genome per milliliter (vg/mL) were determined by qPCR (Applied Biosystems) using SaCas9 or AAT specific primers.

## Serum ALT levels

Alanine aminotransferase (ALT) serum levels were analyzed with a Hitachi Automatic Analyzer (Boehringer Mannheim, Indianapolis, IN).

## Histology and immunohistochemistry (IHC)

Liver sections were fixed with 4% paraformaldehyde (PFA), embedded in paraffin, sectioned (3 μm), and stained with hematoxylin and eosin and were mounted and analyzed by light microscopy for histological evaluation. For IHC all reactions required antigen retrieval, 30 min at 95˚C in 0,01 M Tris-1 mM EDTA pH 9. Incubations with primary antibodies at their optimal dilutions were performed overnight at 4˚C. Primary antibodies used for the IHC were: F4/80 (BioLegend 123102, 1:400), cleaved Caspase-3 (activated Caspase-3 (a-Casp3)) (Cell Signaling 9661, 1:200). For HDAg analysis, patient serum with anti-HDAg reactivity was employed as primary antibody (1:10,000). After rinsing in TBS-T, the sections were incubated with the corresponding secondary antibodies for 30 min at RT. Peroxidase activity was revealed using DAB+ and sections were lightly counterstained with Harris hematoxylin. Finally, slides were dehydrated in graded series of ethanol, cleared in xylene and mounted with Eukitt (Labolan, 28500). Image acquisition was performed on an Aperio CS2 slide scanner using ScanScope Software (Leica Biosystems). All images were stored in uncompressed 24-bit color TIFF format and image analysis was performed using a plugin developed for Fiji, ImageJ (NIH, Bethesda, MD).

## Protein extraction and western blotting

Liver samples were lysed in RIPA buffer (20 mM Tris-HCL pH 7.5, 150 mM NaCl, 1% NaDeoxycholate, 1% Triton X-100 and 1% SDS) supplemented with protease inhibitors. Protein concentration was determined by Pierce BCA Protein Assay Kit (Thermo Scientific, US) according to the manufacturer's recommendations. Western blot was performed using mouse anti-GAPDH (G8795, Sigma-Aldrich, 1:5000), and rabbit anti-RIPK1 (#3493, Cell Signalling, 1:1000) or patient serum with anti-HDAg reactivity, in PBS-T 5% powdered milk. This was followed by incubation with the appropriate horseradish peroxidase (HRP)-conjugated secondary antibody at RT for 1h. SuperSignal West Pico Chemiluminescent Substrate (Thermo Scientific) was used to detect expression.

## On-target mutagenesis analysis

DNA was extracted from liver sections using Macherey-Nagel NucleoSpin Tissue Extraction Kit (Macherey-Nagel, REF 740952.250) with a final elution in 100 μl nuclease-free H2O. A 647 bp genomic region of the targeted locus was enriched by PCR before sequencing. Qubit

quantitation (Invitrogen Qubit dsDNA HS and BR Assays) and TapeStation assay (Agilent Technology D1000 ScreenTape Assay for TapeStation Systems) were employed to quantify and determine the purity of the amplicon. Sequencing libraries were constructed following the NEBNext Ultra II (NEB) library preparation protocol. Sequencing was performed in a MiniSeq (Illumina) instrument with 2x150 paired-end reads at the Cima Genomics Platform (Pamplona, Spain).

The following pipeline was followed for the bioinformatic analysis of indel frequencies: Adapters and low-quality sequences were filtered with Trimmomatic [44]. Pair-end reads were merged with FLASH [45] and mapping to the mouse genome was performed with minimap2 (Li, 2018) with A5 -B4 -O25 -E1 options. Fragments mapping to a window of +/-100 bp around the cutting site were selected with samtools [46] and editing events were quantified with CrispRVariants R package [47,48]. BioProject: PRJNA1090862. Link to the NGS data: https://www.ncbi.nlm.nih.gov/bioproject/PRJNA1090862.

## Statistical analysis

Statistical analysis was performed using GraphPad Prism 7.0 software. The data are presented as mean values ± standard deviation. Differences in ALT levels and % of stained tissue area between groups were analyzed with one-way or two-way ANOVA followed by Bonferroni multiple-comparison test or Kruskal Wallis test followed by Dunns' multiple comparison test. Pearson correlation analysis was performed, and the correlation coefficient calculated. (Significance $*P < 0.05$, $**P < 0.01$, $***P < 0.001$, $****P < 0.0001$).

## Supporting information

**S1 Fig. HDV genomes are primarily detected in hepatocytes, where HDVg and HDVag colocalize, indicating active HDV replication.** Albumin (green) and HDV RNA genome (HDVg in pink) and antigenome (HDVag in yellow) distribution was analyzed by in situ hybridization (ISH) in the liver of C57BL/6 mice 21 days after receiving adenoassociated viral (AAV) vectors delivering both HBV and HDV genomes (HBV/HDV) at a dose of $5x10^{10}$ vg/mouse each. Representative images of hybridized liver sections were captured using the Vectra Polaris Automated Imaging System at various magnifications.
(TIF)

**S2 Fig. HDV genomes are detected in a significant proportion of macrophages.** Albumin (green) and HDVg (pink) and F4/80 (yellow) distribution was analyzed by in situ hybridization (ISH) in the liver of C57BL/6 mice 21 days after receiving adenoassociated viral (AAV) vectors delivering both HBV and HDV genomes (HBV/HDV) at a dose of $5x10^{10}$ vg/mouse each. Representative images of hybridized liver sections were captured using the Vectra Polaris Automated Imaging System at various magnifications. In the last image a macrophage containing HDV RNA has been identify with a circle.
(TIF)

**S3 Fig.** (A) Schematic representation of the recombinant AAV genome carrying the Staphylococcus aureus Cas9 protein flanked by two nuclear localization signals (NLS) and fused to the OLLAS tag under the control of a liver specific promoter TBG (human thyroxine binding globulin promoter) and the guide RNA (sgRNA + gRNA) under the control of U1 promoter. (B) C57BL/6 mice received $10^{11}$ vg of AAV-SaCas9-RIPK1g1 or AAV-SaCas9-RIPK1g2 and 30 hours later animals were sacrificed and RIPK1 expression was analyzed by western blot in liver extracts. (C) Bar chart of allele variant frequency (mean) analyzed in 3 control mice and 3 RIPK1edit mice. Green: no variant, orange: deletion, grey: insertions, pink: others. (D) Indel

size distribution: orange for deletions and grey for insertions. (E) Schematic representation of the experimental procedure, 6/8-week-old C57BL/6 wt mice were iv injected with $10^{11}$ vg of AAV-SaCas9-RIPK1g2 (RIPK1edit) or an AAV expressing SaCas9 without guide (control) and 14 days later animals were challenged with LPS at a dose of 5 μg/gr and sacrificed 6 hours later. Liver damage was analyzed by (F) quantification of serum ALT levels (IU/L) and (G,H) quantification of a-Casp3+ hepatocytes/area after IHC analysis. Statistical analysis was performed by one-way ANOVA followed by Bonferroni multiple-comparison test. $p < 0.05$ (*), $p < 0.0001$ (****), ns = non-significant. (A) Created with BioRender.
(TIF)

**S4 Fig.** (A) Correlation between ALT levels and a-Casp3 positive hepatocytes in wt mice treated as described in Fig 2A. (B-D) C57BL/6 Rag1 KO mice were treated as described in Fig 2A and liver damage was analyzed by (B) quantification of serum ALT levels (U/L) and (C) quantification of a-Casp3+ hepatocytes/area after Immunohistochemistry (IHC) analysis. (D) Correlation between ALT levels and a-Casp3 positive hepatocytes. (E) MLKL expression levels was analyzed in the liver of mice treated daily with a dose of 2.5 mg/kg Nec1 or saline and that were previously injected with AAV-HBV (HBV) or AAV-HBV/HDV (HBV/HDV). Statistical analysis was performed by one-way ANOVA followed by Bonferroni multiple-comparison test. $p < 0.05$ (*), $p < 0.01$ (**), $p < 0.001$ (***), $p < 0.0001$ (****) and ns = non-significant.
(TIF)

**S5 Fig. Macrophage depletion in RIPK1edit and wt mice upon HBV/HDV co-injection.**
(A) IHC analysis against F4/80 was performed at 21 dpi in liver sections of mice receiving saline or clodronate loaded liposomes. Representative images from the different groups are shown. (B) quantitative analysis of F4/80 staining. (C) At sacrifice TNF-α expression was analyzed in the liver of mice by RT-PCR (D, E). The presence of HDV genomes and antigenomes in mouse liver samples was quantified by RT-PCR in CLL treated and untreated animals (D) and in TNF-α KO mice (E). Individual data points and mean values ± standard deviations are shown. Statistical differences were determined by two-way ANOVA.
(TIF)

**S1 Table. Quantitative analysis of the ISH analysis performed in the liver of HBV/HDV mice using HDVg, HDVag and TNF-α RNA probes and DAPI.**
(DOCX)

**S2 Table. Quantitative analysis of the ISH analysis performed in the liver of HBV/HDV mice using HDVg, HDVag and Albumin RNA probes and DAPI.**
(DOCX)

**S3 Table. Quantitative analysis of the ISH analysis performed in the liver of HBV/HDV mice using HDVg, TNF-α RNA probes in combination with anti-F4/80 immunofluorescence and DAPI.**
(DOCX)

**S4 Table. Quantitative analysis of the ISH analysis performed in the liver of HBV/HDV mice using HDVg, Albumin and F4/80 RNA probes and DAPI.**
(DOCX)

**S5 Table. Excell file showing the individual data presented in the graphs of the different figures.**
(XLSX)

## Acknowledgments

We particularly acknowledge the patients for their participation and the Biobank of the University of Navarra for its collaboration. We thank Professor John Taylor, Professor Frank Chisari and Professor Feng Zhang for providing us with essential reagents for our study. We are grateful to Elena Ciordia, Alberto Espinal, and CIFA staff for animal care and vivarium management and to Laura Guembe and Tomas Muñoz from the imaging department at CIMA for their extraordinary technical assistance.

## Author Contributions

**Conceptualization:** Lester Suárez-Amarán, Rafael Aldabe, Gloria Gonzalez-Aseguinolaza.

**Data curation:** Gracián Camps, Sheila Maestro, Laura Torella, Carla Usai.

**Formal analysis:** Sheila Maestro, Laura Torella, Diego Herrero, Carla Usai, Martin Bilbao-Arribas, Ana Aldaz, Rafael Aldabe, Gloria Gonzalez-Aseguinolaza.

**Funding acquisition:** Carla Usai, Gloria Gonzalez-Aseguinolaza.

**Investigation:** Gracián Camps, Sheila Maestro, Laura Torella, Diego Herrero, Carla Usai, Martin Bilbao-Arribas, Rafael Aldabe.

**Methodology:** Gracián Camps, Sheila Maestro, Laura Torella, Diego Herrero, Martin Bilbao-Arribas, Ana Aldaz, Cristina Olagüe, Africa Vales.

**Resources:** Gloria Gonzalez-Aseguinolaza.

**Supervision:** Rafael Aldabe, Gloria Gonzalez-Aseguinolaza.

**Visualization:** Rafael Aldabe.

**Writing – original draft:** Gracián Camps, Sheila Maestro, Laura Torella, Carla Usai.

**Writing – review & editing:** Rafael Aldabe, Gloria Gonzalez-Aseguinolaza.

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
