## [Decision Letter · Decision Letter 0]

21 Dec 2023

Dear Dr Gonzalez-Aseguinolaza,

Thank you very much for submitting your manuscript "Protective Role of RIPK1 Scaffolding against HDV-Induced Hepatocyte cell death and the Significance of Cytokines in Mice" for consideration at PLOS Pathogens. As with all papers reviewed by the journal, your manuscript was reviewed by members of the editorial board and by several independent reviewers. In light of the reviews (below this email), we would like to invite the resubmission of a significantly-revised version that takes into account the reviewers' comments.

Please address all issues raised by each reviewer and note that some of the critiques will required additional experiments, or extensive re-analyses and presentation of available results. Note that both reviewers have raised issues about the image analyses and its pipeline, the efficacy of the RIPK1 knockdown, and the animal experiments.

We cannot make any decision about publication until we have seen the revised manuscript and your response to the reviewers' comments. Your revised manuscript is also likely to be sent to reviewers for further evaluation.

Sincerely,

Luis M Schang, MV. Ph.D.

Guest Editor

PLOS Pathogens

Matthias Schnell

Section Editor

PLOS Pathogens

Kasturi Haldar

Editor-in-Chief

PLOS Pathogens

orcid.org/0000-0001-5065-158X

Michael Malim

Editor-in-Chief

PLOS Pathogens

orcid.org/0000-0002-7699-2064

Please address all issues raised by each reviewer and note that some of the critiques will required additional experiments, or extensive re-analyses and presentation of available results. Note that both reviewers have raised issues about the image analyses and its pipeline, the efficacy of the RIPK1 knockdown, and the animal experiments.

Reviewer's Responses to Questions

**Part I - Summary**

Reviewer #1: The manuscript, "Protective Role of RIPK1 Scaffolding against HDV-Induced Hepatocyte cell death and the Significance of Cytokines in Mice" by Camps et al. provides insights into hepatitis Delta-mediated liver injury in a mouse model. The authors use an AAV-based HBV/HDV infection model in mice to illustrate intra-hepatic pathological changes and the involvement of cytokines and their dependence on RIPK1-mediated signaling.

The present study provides valuable insights for the community. The HBV/HDV mouse model has been described before and has been here applied to study hepatocyte cell death. Overall, the manuscript is of high quality and the conclusion are based on the results.

Specific points to improve the study are listed below:

Reviewer #2: IN this manuscript Camps and colleagues describe their study of the protective role of RIPK1 scaffolding against HDV induced cell death. For this study they employed the AAV-HDV mouse model, which recapitulates many features of HBV/HDV infection in humans. While they have previously shown that HDV induced liver damage can be diminished upon TNF-a treatment they now show that macrophages are the major source of TNF-a during HDV infection. In the current study they show that RIPK1, a downstream molecule of the TNF receptor, partially rescues infected cells from apoptosis via its scaffolding function and not its kinase function. Knock-out or knock-down of RIPK1 resulted in increased HDV-induced liver damage, except in mice that were unresponsive to type-I IFN, indicating that RIPK1 prevented HDV liver damage caused by type-I interferon induced cell death. Moreover they showed that the protective effect of RIPK1 was not linked to TNFa, nor to marcophage activation as depletion of these cells did not impact the event of liver damage.

Overall this is an interesting in vivo study that sheds light on the molecular mechanisms of HDV-induced liver damage, and the protective role RIPK1 plays in this. The study seems well-executed but several aspects could be adjusted to improve the readability of the manuscript (see below).

**Part II – Major Issues: Key Experiments Required for Acceptance**

Reviewer #1: Statistical assessment of image analysis:

It’s unclear how many independent staining’s or animals were used to quantify the number of HDV positive cells or the number of TNFa producing cells in Figure 1 (this apparently also includes the data analysis in other figures). The process is also not particularly well explained in the materials and methods section. The authors should thus provide a broad overview how their image analysis pipeline was set up and how statistical robustness was assessed, including the number of animals used for each experiment.

RIPK1 knockdown

Although a RIPK1 knockdown phenotype is present, the authors should perform sequencing or IF staining to get an idea of the proportion of altered cells in their mouse model. Do the authors also think that overexpression of RIPK1 in this model could provide additional information about its role in HDV pathogenesis? In addition, the authors should provide information on what NEC1 does, as this is not clear from the reading.

Assessment of infectivity

In most of the animal experiments, the authors indicate pathological changes when mice with different genetic background or treatment are infected. However, in most of these experiments, viral load is not quantified. This should be done to evaluate the effects of viral replication in the context of treatment or genotype.

Reviewer #2: Figure 1: the TNFa staining in panel A is not really clear. This picture should be increased in size to make the statement more strong. Panel B, picture 4: HDV antigenomes can be observed in what appears a macrophage. This would suggest that HDV can replicate in macrophages. Is this a rare event? Can this be attributed to macrophages that have engulfed a dying hepatocyte or an apoptotic body originating from a hepatocyte? The text states that 5-10% of TNF-expressing macrophages contain HDV RNA. Is this HDV genome or antigenome? Based on figure 1B-picture 2 and 4 I cannot exclude that the HDV signal is not observed in hepatocytes. A co-staining for a hepatocyte marker could make this clear.

In addition, the authors should include a table with an overview of all cell types (hepatocytes, macrophages, HDV genome positive/negative, HDV antigenome +/-, TNF+/- , ... and their relative presence (%) in the total number of liver cells. Single cell analysis would provide a better view on this, but is perhaps too much/difficult to perform in the context of a revision of this manuscript.

Data related to Figure 2:

- was the effect of ROPK1 KO assessed in mice that only were 'infected' with HDV (without HBV)? Using the AAV model, one would expect that HDV can replicate without the presence of HBV. Also in this case, HDV particles would not be secreted and perhaps not or less be detected/taken up by macrophages?

- panel E: please provide quantitative data, with for HDAg and intracellular genomes (with statistical analysis)

- same panel: is there a potential link between the presence of HDAg and RIPK1 protein (same cells or neighboring cells). A co-staining would elucidate this.

Experiments related to (lack of) involvement of RIPK1 kinase activity:

- Efficacy of New-1 treatment led to significant reduction of MLKL expression levels: please provide fold change.

- P10 2nd paragraph: reference to figure 2E should be figure 2F

Figure 3:

- the overall order of the different panels should be more intuitive (from left to right, and from top to bottom; now a mix).

- ideally this figure starts with a schematic overview in panel A, similarly to figure 2

**Part III – Minor Issues: Editorial and Data Presentation Modifications**

Reviewer #1: Figure 1:

The IF images in Figure 1 were somewhat difficult to interpret because no cell borders or nuclear staining were present. Without these, it is not possible for the reader to quantify the number of cells present in the images. In addition, the total number of cells should also be provided to provide insight into the statistical power of the analysis.

Color code

Some of the colors may be difficult to interpret, especially pink and green when used in the same figure.

Figure 2

The Reference for figure 2E seems to be wrong:

“The efficacy of Nec-1 treatment was evidenced by a significant reduction of MLKL expression levels in the liver of mice (S2E Fig). As shown in Figure 2E, transaminase levels in HBV/HDV injected mice were not affected by Nec-1 treatment.”

Since kinase inhibition with Nec1 does not replicate the knockout phenotype, the authors should either perform additional experiments to clarify how RIPK1 affects HDV infection or discuss this finding in more detail.

Reviewer #2: p6, line 8: "HDV subtle " should be shuttle

PLOS authors have the option to publish the peer review history of their article (what does this mean?). If published, this will include your full peer review and any attached files.

Reviewer #1: No

Reviewer #2: No
---

## [Decision Letter · Decision Letter 1]

16 Apr 2024

Dear Dr Gonzalez-Aseguinolaza,

We are pleased to inform you that your manuscript 'Protective Role of RIPK1 Scaffolding against HDV-Induced Hepatocyte cell death and the Significance of Cytokines in Mice' has been provisionally accepted for publication in PLOS Pathogens.

Best regards,

Luis M Schang, MV. Ph.D.

Guest Editor

PLOS Pathogens

Matthias Schnell

Section Editor

PLOS Pathogens

Michael Malim

Editor-in-Chief

PLOS Pathogens

orcid.org/0000-0002-7699-2064

Thank you for submitting a highly revised manuscript with ample new data and analyses and clear explanations of the technical reasons why some of the critiques could not be addressed experimentally.

Reviewer Comments (if any, and for reference):

Reviewer's Responses to Questions

**Part I - Summary**

Reviewer #1: All points sufficiently addressed. Thank you.

**Part II – Major Issues: Key Experiments Required for Acceptance**

Reviewer #1: (No Response)

**Part III – Minor Issues: Editorial and Data Presentation Modifications**

Reviewer #1: (No Response)

PLOS authors have the option to publish the peer review history of their article (what does this mean?). If published, this will include your full peer review and any attached files.

Reviewer #1: No

---

## [Editor Report · Acceptance letter]

6 May 2024

Dear Dr Gonzalez-Aseguinolaza,

We are delighted to inform you that your manuscript, "Protective Role of RIPK1 Scaffolding against HDV-Induced Hepatocyte cell death and the Significance of Cytokines in Mice," has been formally accepted for publication in PLOS Pathogens.

Best regards,

Michael Malim

Editor-in-Chief

PLOS Pathogens

orcid.org/0000-0002-7699-2064